# Toll-Like Receptor Response to Hepatitis B Virus Infection and Potential of TLR Agonists as Immunomodulators for Treating Chronic Hepatitis B: An Overview

**DOI:** 10.3390/ijms221910462

**Published:** 2021-09-28

**Authors:** Mohammad Enamul Hoque Kayesh, Michinori Kohara, Kyoko Tsukiyama-Kohara

**Affiliations:** 1Transboundary Animal Diseases Centre, Joint Faculty of Veterinary Medicine, Kagoshima University, Kagoshima 890-0065, Japan; mehkayesh@pstu.ac.bd; 2Department of Microbiology and Public Health, Faculty of Animal Science and Veterinary Medicine, Patuakhali Science and Technology University, Barishal 8210, Bangladesh; 3Department of Microbiology and Cell Biology, Tokyo Metropolitan Institute of Medical Science, Tokyo 156-8506, Japan; kohara-mc@igakuken.or.jp

**Keywords:** hepatitis B virus, chronic, infection, toll-like receptor, TLR agonists

## Abstract

Chronic hepatitis B virus (HBV) infection remains a major global health problem. The immunopathology of the disease, especially the interplay between HBV and host innate immunity, is poorly understood. Moreover, inconsistent literature on HBV and host innate immunity has led to controversies. However, recently, there has been an increase in the number of studies that have highlighted the link between innate immune responses, including Toll-like receptors (TLRs), and chronic HBV infection. TLRs are the key sensing molecules that detect pathogen-associated molecular patterns and regulate the induction of pro- and anti-inflammatory cytokines, thereby shaping the adaptive immunity. The suppression of TLR response has been reported in patients with chronic hepatitis B (CHB), as well as in other models, including tree shrews, suggesting an association of TLR response in HBV chronicity. Additionally, TLR agonists have been reported to improve the host innate immune response against HBV infection, highlighting the potential of these agonists as immunomodulators for enhancing CHB treatment. In this study, we discuss the current understanding of host innate immune responses during HBV infection, particularly focusing on the TLR response and TLR agonists as immunomodulators.

## 1. Introduction

Hepatitis B virus (HBV) infection, which causes chronic hepatitis, liver cirrhosis, and hepatocellular carcinoma (HCC), remains a major global health problem despite the availability of an effective preventive vaccine for HBV [1,2]. According to a recent estimate, 296 million people were affected by chronic HBV infection in 2019, with an annual 1.5 million new infections [3]. The current treatment involves administration of pegylated interferon alpha (Peg-IFN-α) and nucleos(t)ide analogs (NAs), including three nucleoside analogs (lamivudine, entecavir, and telbivudine) and three nucleotide analogs (adefovir, tenofovir disoproxil, and tenofovir alafenamide). However, these drugs can only suppress HBV replication; they cannot completely eliminate HBV infection, mostly because of the persistent nature of the covalently closed circular DNA (cccDNA) and/or integrated HBV DNA in hepatocytes [4]. HBV is an enveloped, circular, and partially double-stranded relaxed circular DNA (rcDNA) virus [5]. HBV belongs to the family *Hepadnaviridae*, which contains a genome of approximately 3.2 kb with four overlapping open reading frames, encoding seven proteins, including polymerase, core, precore, three envelope/surface proteins (large, middle, and small), and X protein [6,7,8]. The envelope and core proteins are structural proteins, whereas X and polymerase are non-structural proteins [8]. Sodium taurocholate co-transporting polypeptide (NTCP) has been demonstrated as an entry receptor for HBV [9], and the interaction is host-specific [10]. A complex series of events occurs between a virus and the host immune system, which determines the outcome of an infection [11]. Similarly, the outcome of HBV infection largely depends on the virus–host interactions, where, in 95% of the immunocompetent adults the infection is cleared, but in over 90% of the infected neonates this fails and they develop chronic infection [12,13]. Chronic HBV infection is detected by the continuous expression of hepatitis B surface antigen (HBsAg) for at least 6 months after the initial infection. There are inconsistent data on HBV–host interactions, particularly innate immune responses. For example, the lack of innate immune response, including induction of interferons (IFNs) and IFN-stimulated genes (ISGs), in acute HBV-infected chimpanzees [14] the woodchuck model of HBV infection [15] and patients with HBV infection [16] highlights HBV as a stealth virus. A recent study demonstrated that HBV remains invisible to pattern recognition receptors (PRRs) [17], which could be due to the ability of HBV proteins to inhibit or evade the host innate immune system [18,19,20,21,22]. Notably, in our previous study, we observed a significant suppression of interferon regulatory factor 7 (IRF-7) and ISG15, as well as no increase in IFN-β production at 1- or 3-days post-infection in the tree shrew model [23], which was suggestive of the inhibition of innate immune response at early stage of infection in this model. However, it is usually difficult to identify the exact time-point of initial infection of the host, which makes it difficult to characterize the early stages of HBV infection. Moreover, the differences in innate immune response, which may be due to the differences in HBV genotype, infection state, or host genetics, remain largely undefined. Nevertheless, numerous recent studies have indicated the sensing of HBV by PRRs, including Toll-like receptors (TLRs) [24,25]. The innate immune response plays an important role as the first line of immune defense against many viral infections [26]. TLRs, retinoic acid-inducible gene I (RIG-I)-like receptors (RLRs), nucleotide-binding oligomerization domain (NOD)-like receptors (NLRs), and C-type lectin receptors play an essential role in sensing the invading pathogens, including viruses, and initiating an innate immune response. This leads to the synthesis of IFNs and cytokines through several distinct signaling pathways, thereby limiting infection and promoting adaptive immune responses [27,28]. Although the role of adaptive immunity in controlling HBV infection is well documented, the role of innate immunity in this regard is yet to be largely explored [18]. Moreover, the mechanisms behind HBV-specific immune responses [29] remain to be investigated, which may pave the way for development of new strategies for the treatment of chronic HBV infection. TLRs are evolutionarily conserved key molecules of innate immunity, which are involved in the early detection of invading microorganisms by sensing pathogen-associated molecular patterns (PAMPs) [30,31,32]. TLRs are type I transmembrane proteins containing three domains: an extracellular leucine-rich repeat domain that recognizes specific PAMPs, a single transmembrane domain, and an intracellular Toll-interleukin 1 receptor domain required for downstream signal transduction [33,34]. TLR signaling is involved in the regulation of both pro- and anti-inflammatory cytokines and links the early innate and adaptive immunity [35]. Several studies have indicated the association of TLR-mediated signaling with antiviral mechanisms and suppression of HBV replication [36,37], as well as TLR response suppression with HBV persistence [38]. Therefore, activation of innate immune response by TLR agonists may play a significant role in modulating the outcome of chronic HBV infection. Moreover, TLRs are important triggering molecules in activating trained immunity and can be used as vaccine adjuvants [39,40,41,42]. In a previous study, the phenomenon of trained immunity in newborn infants of HBV-infected mothers was reported [43]. Thus, the use of TLR agonists as HBV-specific therapeutics/vaccines could be investigated to enhance the immune response against chronic infections. In this study, we have discussed our current understanding of HBV–host interactions by focusing on TLR–HBV interactions during chronic infections in human and animal models. In addition, we have also addressed the immunomodulatory potential of TLR agonists for improving host innate immune responses during chronic HBV infection.

## 2. TLR Response to HBV Infection

Although innate immune response during the early phase of HBV infection is considered to be negligible, an increasing number of studies have indicated that HBV infection modulates innate immune responses, including TLR response [38]. HBV has a restricted host range, naturally infecting humans and chimpanzees, and to some extent, tree shrews [44,45,46], which also limits the early immune response study of HBV infection. In a previous study, Isogawa et al. reported the induction of IFN-α/β in the liver of HBV-transgenic mice within 24 h of a single intravenous injection of TLR3, 4, 5, 7, and 9 agonists, which inhibited HBV replication [37], suggesting the anti-HBV role of these TLRs. TLR-mediated control of HBV replication using TLR agonists has also been demonstrated in mice non-parenchymal liver cells [47]. Markedly low expression of TLR transcripts, including TLR1, 2, 4, and 6, was observed in peripheral blood mononuclear cells (PBMCs) obtained from patients with chronic hepatitis B (CHB). The cells also showed an impaired cytokine response after stimulation with TLR2 and TLR4 agonists, which correlated with the plasma HBsAg level of the patients [48], suggesting a possible interaction between HBsAg and TLR signaling. Another study showed that after stimulation with ligands for TLR2, TLR3, and TLR9 on PBMCs obtained from children with chronic HBV infection, there was an increase in production of IL-6, CCL3, and CXCL10 [49], indicating the induction of TLR-mediated inflammatory response. However, on stimulation with ligands for TLR2, TLR3, and TLR4, the PBMCs from children with CHB showed a significantly lower IFN-α production than in those from healthy children [49], suggesting a suppressed IFN response. A significantly decreased expression of TLR signaling molecules, including IRAK4, TRAF3, and IRF7, was found in PBMCs of patients with CHB, as compared to that in those of healthy controls [50], suggesting an impaired immune response in chronic HBV infection. Genetic variations in the *TLR3* gene were found to affect the outcome of HBV infection [51,52], highlighting the involvement of TLR3. Decreased expression of both TLR7 mRNA and protein was observed in PBMCs obtained from patients with CHB, as compared to healthy controls; however, a decreased *TLR9* mRNA expression but an increased TLR9 protein level was observed in patients with CHB, which correlated with their serum HBV DNA, suggesting a possible link between TLR9 protein expression and HBV replication [53]. In another study, the expression of both TLR9 mRNA and protein was found to be downregulated in PBMCs from patients with CHB [54], highlighting the association of TLR9 in HBV infection. A previous study reported a lowered production of IFN-α by pDCs from patients with CHB, which was in response to loxoribine, a ligand for TLR7, and CpG ODN, a ligand for TLR9 [55]. Moreover, several studies have reported an inverse correlation between the number of pDCs and the expression of TLR9 in pDCs with the serum HBV load [55,56], suggesting the antiviral role of TLR9 in HBV infection. A previous study reported the suppression of TLR9-induced IFN-α production by plasmacytoid DCs obtained from HBeAg-positive patients with CHB [57]. In addition, a recent study indicated the predictive value of pDCs and TLR9 in HBeAg-positive patients with CHB; after IFN-α treatment, pDC-mediated expression of blood dendritic cell antigen 2 (BDCA-2), and expression of immunoglobulin-like transcript 7 (*ILT7*) and *TLR9* mRNA were significantly increased in the response group compared with that in the non-responders group [58].

Similar to that of patients with CHB, the transcriptomic analysis of the woodchuck model of CHB also revealed a limited intrahepatic type I IFN response [59]. In the woodchuck model of chronic hepatitis, a suppressive trend of TLR expression was reported in the hepatocytes, compared to that in healthy animals, suggesting an impairment of the innate immune response in chronic infection [60]. However, a recent study indicated the role of TLR2 in the resolution of HBV infection in a woodchuck model of hepatitis [61]. In our previous study, we found a significant suppression of IFN-β response at 31 weeks post-infection in the tree shrew model [62], which might have contributed to the chronicity. Moreover, TLR3 was not induced and TLR9 was suppressed [62]. Notably, a previous study also reported decreased expression of TLR9 in peripheral CD14+ monocytes collected from patients with CHB [63]. Although the mechanism remains unknown, a previous study reported that responders to pegylated IFN and those under ETV treatment showed restoration of TLR9 expression, suggesting the role of TLR9 in HBV inhibition [63]. Similarly, an earlier study demonstrated a reduction in TLR3 expression in PBMCs and the liver cells of patients with CHB compared to healthy controls, which was restored by IFN therapy, suggesting the role of TLR3 in HBV inhibition [64]. TLR2 and 4 may inhibit HBV replication in an IFN-independent manner by activating MAPK and PI-3 K/Akt pathways in hepatocytes [36]. Notably, a recent study showed the sensing of HBV and induction of anti-HBV immune response through TLR2 signaling after infection in primary human hepatocytes (PHH) [25], a verified in vitro model of HBV infection [65]. Suppressed expression of TLR2 has been observed in chronic HBV infections [66,67], with significantly decreased expression of TNF-α and IL-6 production [66]. In addition, a recent study demonstrated the association of TLR2 in the regulation of T helper 17 (Th17) cell response in HBV infection [68]. Using HBV mouse model, other recent studies have demonstrated the role of TLR signaling in enhancing HBV-specific CD8+ T-cell responses, which controls the infection [25,69]. Another study reported a reduced expression of TLR7 in HBV-replicating HepG2.2.15 cells and in the liver biopsy samples from patients with CHB; an inverse relationship between HBV DNA load and TLR7 expression in biopsy samples was observed [70], highlighting the antiviral role of TLR7 in HBV infection, which was further confirmed by the suppression of HBV replication in HepG2.2.15 cells by TLR7 agonist, R837 [70]. In HBV-transgenic mice, saturated fatty acids (SFAs), potential ligands for TLR4, were shown to accelerate TLR4 signaling, which inhibited HBV replication in CHB infection with non-alcoholic fatty liver disease [71], suggesting an antiviral role of TLR4 against HBV infection. The role of the innate immune response in inhibiting HBV replication has also been shown in an HBV hydrodynamic mouse model, where an intraperitoneal inoculation of STING agonist, 5,6-dimethylxanthenone-4-acetic acid (DMXAA), induced type I IFNs that reduced HBV DNA replication intermediates in the mouse livers [72]. In mouse hepatocytes supporting HBV replication, the activation of STING with either cGAMP or STING agonist significantly reduced the viral DNA in a STING- and Janus kinase 1-dependent manner [73]. Overall, these findings suggest that innate immunity plays a significant role in the inhibition of HBV replication; however, further studies are required to explore the mechanisms. The interaction of TLRs in different models of HBV infection is shown in Figure 1 (Figure 1A–F), which suggests the crucial role of TLR response in inhibiting HBV infection. In this regard, we also have shown the association of TLRs with other molecules, including IRAK4, TRAF3, and IRF7 in the induction of IL-6, CCL3, and CXCL10 [36,49,74] (Figure 1G). Therefore, a proper understanding of TLR responses in HBV infection is critical for successful therapeutic or preventive interventions. Also, potentiality of TLR agonists, particularly agonists for TLR7, 8, and 9 are under investigation in several clinical trials, which appear promising as immunomodulators to enhance host immune response [74], will be discussed in the later part of the study.

## 3. Inhibition of Innate Immune Response by HBV Infection

The exact mechanisms of immune evasion or inhibition by HBV still remain unclear. Moreover, whether HBV evades or inhibits innate recognition, or stimulates innate immunity has not yet been verified [75]. However, HBV protein levels, including HBsAg and HBeAg, are associated with HBV persistence, which is suggestive of the role of viral proteins in the suppression of host immune response [76,77,78]. Several studies have enhanced our understanding of the innate immune response modulation by HBV [38,79]. HBV has been reported to suppress TLR-mediated antiviral response in hepatic cells by interfering with the activation of IRF-3, nuclear factor kappa B (NF-κB), and extracellular signal-regulated kinase (ERK) 1/2 [80]. HBV has been shown to inhibit TLR9 response by inhibiting the MyD88-IRAK4 axis in pDCs obtained from patients [54]. In another study, TLR9 expression and TLR9-mediated B cell functions were suppressed in all peripheral B cell subsets exposed to HBV [81]. Impaired expression of TLR4, 8, and 9 in peripheral DC subsets from patients with chronic HBV infection has also been reported [82], which induced a reduced innate immune function. HBV polymerase was reported to block IRF activation by disrupting the interaction between IκB kinase-ε and DEAD box RNA helicase and inhibiting IFN production [83]. Another study showed the inhibition of STING-mediated IRF-3 activation and IFN-β production by HBV polymerase [84]. HBsAg has been reported to suppress the activation of NF-κB, IRF-3, and MAPKs in murine hepatocytes [85]. Moreover, T-cell activation induced by TLR3-stimulated murine KCs or LSECs was also suppressed by HBsAg [85], suggesting the potential of HBsAg to attenuate or inhibit TLR-mediated immune response. Using RPMI 8226, a human myeloma B cell line, it was shown that HBsAg induced TLR9 dysfunction by suppressing HBsAg-induced phosphorylation, which activated the transcription factor, CREB, thereby preventing TLR9 promoter activity [81]. A recent study also reported HBsAg-mediated interference of the NF-κB pathway by interaction with TAK1 and TAB2, leading to a suppressed immune response [86]. Moreover, HBc-mediated downregulation of interferon-induced transmembrane protein 1 (IFITM1) expression [87] and HBeAg-mediated suppression of NF-κB signaling pathway, inhibiting the innate immune response [88,89], has also been reported. The HBV X protein (HBx) plays a critical role in inhibiting the innate immune response, and several studies have indicated the association of HBx in the suppression of type I IFN production by downregulating MAVS protein [90,91,92,93]. HBx-mediated downregulation of TIR-domain-containing adaptor inducing interferon-beta (TRIF), a key component of the innate immune signaling, has also been shown [43]. HBx can suppress the transcription of TRIM22 through a single CpG methylation in its 5’-UTR, reducing the binding affinity of IRF-1 and inhibiting IFN-mediated anti-HBV response [94]. Recently, a key mechanism of innate immune evasion by HBx in hepatocytes has been reported, where HBx-induced adenosine deaminases acting on RNA 1 (ADAR1) deaminates adenosine to generate inosine by interacting with HBV RNAs and avoiding host immune recognition [19]. Immune evasion by attenuating RIG-I signaling via N^6^ methyladenosine (m^6^A) modification of HBV RNAs has been reported previously [95]. Recently, HBV-induced N^6^ m^6^A modification of phosphatase and tensin homolog (PTEN) affecting innate immunity with decreased IRF-3 dephosphorylation and IFN synthesis has also been reported [96]. Zhou et al. reported that HBV inhibited RIG-I-induced IFN production by forming a ternary complex including hexokinase and sequestering MAVS from RIG-I [97]. Impairment of NK cell function by HBV has also been reported [98]. HBV can evade cGAS sensing in HepG2-NTCP cells with a fully functional cGAS-STING pathway [99]; however, the mechanism remains unknown. Moreover, the demonstration of the sensing of naked HBV rcDNA but not the infectious HBV virions by cGAS in PHH indicates the impairment of cGAS sensing during HBV infection [99,100], although the exact mechanism remains unclear. From the above findings, it is reasonable to consider that establishing HBV infection and its protein components can inhibit/suppress the host innate immune response (Figure 2) by different known and unknown mechanisms, and therefore, a detailed understanding of the immune inhibition/evasion is crucial for devising new therapeutic and preventive strategies to control HBV infection.

## 4. Potential of TLR Agonists as Immunomodulators

As immune response is suppressed in chronic HBV infection, restoration of the immune response is crucial for improving the chronic infection outcome, which can be achieved using immunomodulators, such as TLR agonists, checkpoint inhibitors, and therapeutic vaccines [4,101,102]. In this study, we mainly focused on the use of TLR agonists as immunomodulators to enhance the immune response in chronic infection. TLR agonists play a significant role in modulating the immunotherapeutic effects [103,104]. Recently, TLR agonists have attracted interest for use as vaccine adjuvants or immune modulators because of their ability to induce the production of IFN, proinflammatory cytokines and chemokines, which may exert anti-HBV effects [105,106]. TLR1/2 and TLR3 agonists can inhibit HBV replication in PHH [105]. Another study showed that GS-9620 (vesatolimod), an oral TLR7 agonist, along with nucleos(t)ide analogs (NAs) positively enhanced the immunomodulatory effects by increasing the T cell and NK cell responses and reducing the ability of NK cells to suppress T cells in chronically infected patients [107]. However, no significant change in HBsAg level was found after combination therapy with NAs and GS-9620 [107]. A recent study demonstrated that dual-acting TLR7/8 (R848) and TLR2/7 (CL413) agonists are more potent in inhibiting HBV replication than single-acting TLR7 (CL264) or TLR9 (CpG-B) agonists [108], which highlights the higher potential of dual-acting TLR agonists in inducing broad cytokine repertoires. Several studies have demonstrated the ability of TLR agonists to suppress HBV in animal models. In a previous study, GS-9620 was shown to induce an immune response by increasing the production of IFN-α, other cytokines, and chemokines in chronically infected chimpanzees. It reduced viral loads over 2 logs and serum HBsAg and HBeAg levels by 50% [109], highlighting the effectiveness of TLR7 agonists in the treatment of chronic HBV infection and supporting the further investigation of this candidate in human CHB. In another recent study, the mechanism of GS-9620 action was investigated in CHB chimpanzees, and it was reported that GS-9620 exerting anti-HBV response associated with aggregation of immune cells in the liver that can either kill HBV-infected cells or can prevent HBV from infecting new cells by producing antibodies [110]. In the woodchuck model of CHB, the administration of the GS-9620 also induced a significant reduction in serum viral DNA and intrahepatic woodchuck hepatitis virus (WHV) DNA replicative intermediates, WHV cccDNA and WHV RNA [111], indicating that the TLR7 agonist is a potent immunomodulator with anti-HBV effects in the woodchuck model. Moreover, the incidence of HCC was remarkably reduced in GS-9620-treated woodchucks [111]. In another study, it was shown that oral administration of another TLR7 agonist, APR002, in combination with entecavir (ETV) enhanced ISG expression and reduced WHV cccDNA in chronically infected woodchucks with WHV [112]. In a phase 1b clinical trial, oral administration of GS-9620 was found to be safe and well-tolerated, which enhanced peripheral ISG15 production without significant systemic IFN-α levels [113]. AL-034, an oral TLR7 agonist, showed efficacy against HBV in a mouse model [114]. The evaluation of HBV therapeutic vaccine that consisted of a novel TLR7 agonist (named as T7-EA), an alum adjuvant, and a recombinant HBsAg protein, indicated that T7-EA induced Th1-type immune responses, as well as increased T-cell response and HBsAg-specific IgG2a titer in a mouse model [115]. An earlier study reported that TLR8 agonist ssRNA40 could selectively activate liver-resident innate immune cells, and mucosal-associated invariant T cells and NK cells were identified as IFN-γ-producing cells after TLR8 activation [116], which highlighted the possibility of using TLR8 agonist in treating CHB. GS-9688 (selgantolimod), an oral TLR8 agonist, was found as a potent anti-HBV molecule in HBV-infected primary human hepatocytes [117]. In another study, GS-9688 was reported to exert anti-HBV effect in the woodchuck model of CHB [118], where it reduced viral loads over 5 logs and suppressed woodchuck hepatitis surface antigen (WHsAg) in 50% of the treated chronically infected woodchucks [118]. Another recent study also showed the therapeutic potential of GS-9688 in CHB, where GS-9688 induced cytokines in human PBMCs that could activate antiviral effector function by different immune mediators, including NK cells, HBV-specific CD8+ T cells, CD4+ follicular helper T cells, and mucosal-associated invariant T cells [119]. The anti-HBV activity of TLR9-ligand has been reported previously [120,121]. The TLR9 agonist, CpG oligodeoxynucleotides (ODNs), was also evaluated in the woodchuck model, and it was observed that the WHsAg serum level was suppressed by the combined administration of CpG and ETV, but no effect was observed with either agent alone [122], suggesting synergistic effects. CpG oligonucleotides increased HBV-specific IL2 and IFN-γ responses in whole blood stimulated with HBsAg or HBcAg [123]. An earlier study reported the ability of AIC649, TLR9 agonist, in reducing WHV DNA, and WHsAg in a unique biphasic response pattern [124]. Recently, it was shown that AIC649, in combination with ETV, could effectively suppress WHV DNA and WHsAg in the woodchuck model of CHB [125]. A previous study showed the use of TLR3 ligand, poly (I:C), as an effective adjuvant for HBV therapeutic vaccine (named as pHBV-vaccine), which effectively suppressed HBV replication [126]. A recent study also reported the improvement of liver infection status when poly (I:C) was delivered by calcium phosphate nanoparticles conjugated with an F4/80 antibody. It also enhanced T cell responses and production of intrahepatic cytokines and chemokines, which significantly reduced HBsAg, HBeAg, and HBV DNA levels in mice [127]. A recent study showed that HBV interacted with SIGLEC-3 (CD33) and served as an immune checkpoint receptor for HBV infection and impaired the host immunity [128]. In PBMCs from CHB patients, it has also been shown that anti–SIGLEC-3 mAb could reverse the effect and induce a cytokine response to the TLR-7 agonist, GS-9620 [128]. TLR agonists as vaccine adjuvants are currently under investigation for different human vaccines, including HBV vaccines, and appear promising in vaccine studies [129,130]. Vaccines containing particular TLR agonist(s) may activate specific TLR(s) and enhance vaccine efficacy without direct participation in protective immunity [131,132]. In an investigation of a therapeutic synthetic long peptide (SLP)-based vaccine to treat chronic HBV, it has been shown that TLR2-ligand conjugation of the prototype HBV-core SLP triggered functional patient T cell responses ex vivo [133], suggesting that TLR agonists may also act as potential adjuvants in HBV vaccines. A recent study has also demonstrated the ability of PRR ligands to induce innate immunity toward HBV control [134]. Therefore, the use of TLR agonists in HBV therapeutics/vaccines seems promising and could be an effective tool in the control of HBV chronic infection, which requires further investigation. The TLR agonists as immunomodulators in development are shown in Table 1.

## 5. Conclusions

Despite the availability of an effective HBV vaccine, chronic HBV infection remains a global health problem. Recent studies have highlighted HBV as a cunning virus, which also raises questions about stealth properties of HBV and shows the ability of the virus to interfere with the innate immune response and establish an infection. From the available data, it is assumed that chronic HBV infection induces the dysregulation of host innate immunity, including the suppression of TLR response and downstream cytokines. The induction of host innate immune response by using TLR agonists may aid in the better understanding of the host innate immune responses, particularly the interaction between TLRs and viral components. Thus, although TLR agonists have shown promising results in improving the innate immune response during HBV infection, further studies are required to investigate the mechanisms behind modulating the host immune response and restoring the immunity.

## Figures and Tables

**Figure 1 ijms-22-10462-f001:**
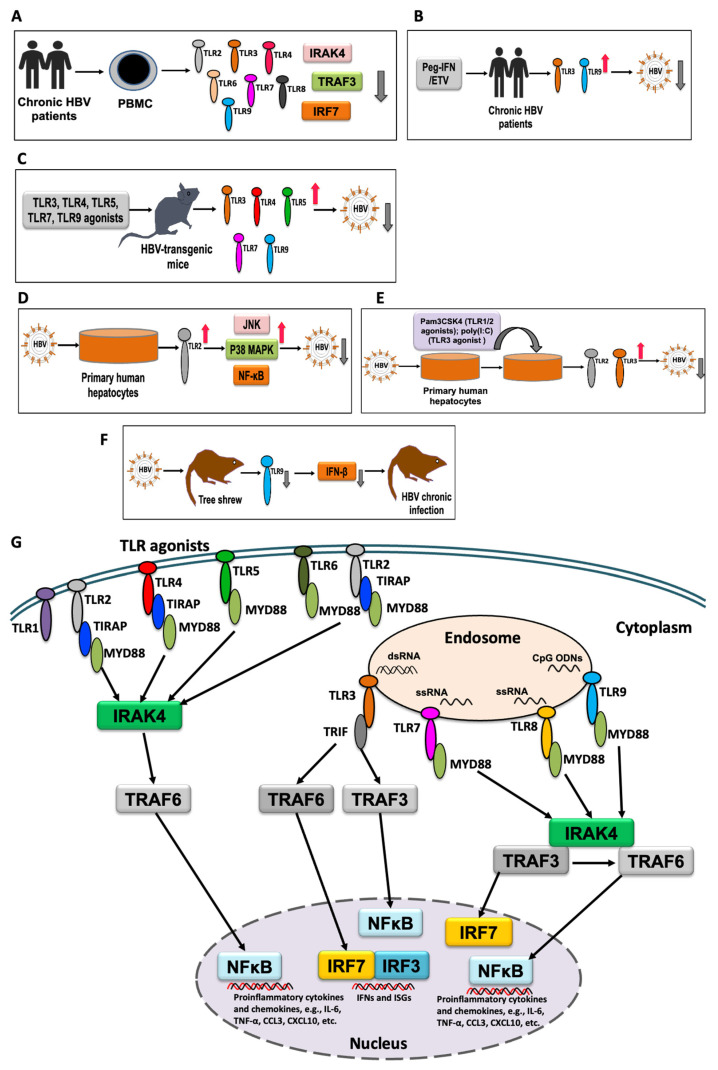
An overview of TLR response in different models of HBV infection. (**A**) Suppression of TLRs and its adaptor components in patients with chronic HBV infection. (**B**) Restoration and anti-HBV response of TLRs in patients with chronic HBV infection after Peg-IFN or ETV therapy. (**C**) Induction of TLR responses by using TLR agonists and inhibition of HBV replication in HBV-transgenic mice. (**D**) Inhibition of HBV by TLR2 and downstream signaling components in HBV-infected primary human hepatocytes (PHH). (**E**) Anti-HBV response of TLR2 and TLR3 after using TLR1/2 and TLR3 agonists in HBV-infected PHH. (**F**) Suppression of TLR9 expression and downstream cytokine during chronic HBV infection in tree shrew model. (**G**) Cartoon showing the relationship of TLRs with other signaling molecules, including IRAK4, TRAF3, and IRF7 in the induction of IL-6, TNF-α, CCL3, and CXCL10. Upon activation of TLRs by respective TLR ligands, adaptor molecules are recruited, and through downstream signaling pathways proinflammatory cytokines, chemokines, IFNs, and ISGs are activated and produced.

**Figure 2 ijms-22-10462-f002:**
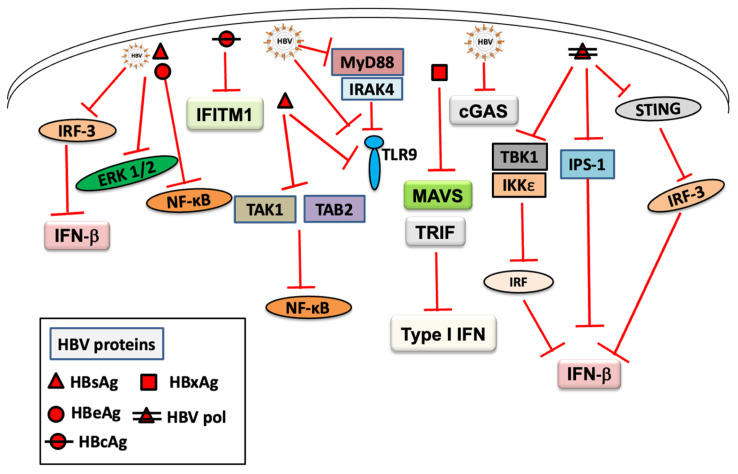
An overview of the mechanism of the inhibition of host innate immune response by HBV and its proteins.

**Table 1 ijms-22-10462-t001:** TLR agonists as immunomodulators for targeting HBV.

Compound (Company)	Target	Clinical Phase of Development	Effects on Host Immunity	Effects on HBV	References
GS-9620 (Gilead Sciences)	TLR7 agonist	Phase II	Increased T-cell and NK cell response; dose-dependent ISG15 induction; no increase of systemic IFN-α level	No decrease of serum HBsAg level	[107,113,135,136]
RO7020531 (Hoffmann-La Roche)	TLR7 agonist	Phase I	Safe and acceptably tolerated in healthy volunteer	No clinical data available	[137]
JNJ-64794964 (Janssen)	TLR7 agonist	Phase I	Safe and well tolerated in healthy volunteer	No clinical data available	[138]
GS-9688 (Gilead Sciences)	TLR8 agonist	Phase II	Dose-dependent cytokine induction	Decrease of HBsAg and HBeAg level by 24 weeks	[139]
AIC649 (AiCuris)	TLR9 agonist	Phase I	Increase in IL-1β, IL-6, IL-8 and IFN-γ and reduction in IL-10 levels	No substantial change in HBV DNA and HBsAg level	[140]

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
