# Peer review of "Toll-Like Receptor Response to Hepatitis B Virus Infection and Potential of TLR Agonists as Immunomodulators for Treating Chronic Hepatitis B: An Overview"

_ijms, 2021, doi:10.3390/ijms221910462_

Round 1

Reviewer 1 Report

The aim of this manuscript is to discuss the current understanding of HBV-host interactions, by highlighting Toll- like receptors and HBV interactions in chronic infections, in human and animal models.

Even if the review presents a densely organized structure and is based on well-synthetized data, there are aspects to be mentioned, to make the mauscript fully readable. For these reasons, the article requires minor changes.

Please find below an enumerated list of comments on my review of the manuscript:

LINE 32: Hepatitis B Virus infection is a global health problem and a major cause of acute anche chronic liver disease. This issue is analyzed by recent studies (see, for reference: Long-term immune protection against HBV: associated factors and determinants – 2021), which also highlight the importance of a long-term immune protection and the immonogenicity of HBV vaccine.

LINE 279: The antiviral response to GS-9620 is also associated to aggregation of immune cells in the liver, that can either kill cells, infected with HBV or enhance the production of antibodies, in order to avoid the infection of new liver cells. This issue is higlighted by a recent study, also conducted on chronically infected chimpanzees (see, for reference: Anti-HBV response to toll-like receptor 7 agonist GS-9620 is associated with intrahepatic aggregates of T cells and B cells – 2018).

To sum up, the topic is timely and call for attention. Overall, the review requires minor changes (as mentioned). I would accept the manuscript, if the comments are addressed properly.

Author Response

LINE 32: Hepatitis B Virus infection is a global health problem and a major cause of acute anche chronic liver disease. This issue is analyzed by recent studies (see, for reference: Long-term immune protection against HBV: associated factors and determinants – 2021), which also highlight the importance of a long-term immune protection and the immonogenicity of HBV vaccine.

Response: We would like to thank the reviewer for his comments. Accordingly, we have added this reference and relevant information (page 1, line 32-33).

LINE 279: The antiviral response to GS-9620 is also associated to aggregation of immune cells in the liver, that can either kill cells, infected with HBV or enhance the production of antibodies, in order to avoid the infection of new liver cells. This issue is higlighted by a recent study, also conducted on chronically infected chimpanzees (see, for reference: Anti-HBV response to toll-like receptor 7 agonist GS-9620 is associated with intrahepatic aggregates of T cells and B cells – 2018).

Response: We would like to thank the reviewer for pointing out this recent issue. In line up with the reviewer comments we have updated the text including the suggested study (page 7, line 292-296).

To sum up, the topic is timely and call for attention. Overall, the review requires minor changes (as mentioned). I would accept the manuscript, if the comments are addressed properly.

Response: We would like to thank the reviewer for his sincere comments and kind reading of the manuscript. We hope that we have duly addressed the issues pointed by the reviewer.

Reviewer 2 Report

HBV infection is an important issue, especially in Asia. The authors discuss the relation between TLR and HBV.  
Some questions that need to answer.
Q1: In the section on TLR response to HBV infection, the author described the association of TLR 2, 3, 9 with HBV, but the clinical trait focus on the TLR 7, 8, 9 [Gehring AJ, Protzer U: Targeting Innate and Adaptive Immune Responses to Cure Chronic HBV Infection. Gastroenterology 2019;156:325- 337]. Should authors discuss more TLR 7, 8, and 9?
Q2: Authors can draw a picture to show the relationship of TLR with other molecules, like IL-6, CCL3, and CXCL10 or IRAK4, TRAF3, and IRF7.
Q3: In the section on the Potential of TLR agonists as immunomodulators, authors can list a table about TLR agonists, targets, and functions. 

Author Response

Q1: In the section on TLR response to HBV infection, the author described the association of TLR 2, 3, 9 with HBV, but the clinical trait focus on the TLR 7, 8, 9 [Gehring AJ, Protzer U: Targeting Innate and Adaptive Immune Responses to Cure Chronic HBV Infection. Gastroenterology 2019;156:325- 337]. Should authors discuss more TLR 7, 8, and 9?

Response: We would like thank the reviewer for his/her valuable comments. In line with the reviewer comments, we have discussed more the commented point (page 8, line 292-296; page 8, line 306-307; 310-323; 328-331).

Q2: Authors can draw a picture to show the relationship of TLR with other molecules, like IL-6, CCL3, and CXCL10 or IRAK4, TRAF3, and IRF7.

Response: In line with reviewer comment we have included a picture showing the relationship of TLR with other molecules (newly added figure 1G).

Q3: In the section on the Potential of TLR agonists as immunomodulators, authors can list a table about TLR agonists, targets, and functions.

Response: In response to reviewer comment we have added a table (newly added Table 1) listing the potential TLR agonists, their targets and functions.

Round 2

Reviewer 2 Report

The authors answer all the questions and the manuscript can be accepted.